# Generation and Characterization of a Replication-Competent Human Adenovirus Type 55 Encoding EGFP

**DOI:** 10.3390/v15051192

**Published:** 2023-05-18

**Authors:** Wei Li, Yuehong Chen, Ye Feng, Jing Li, Xiaoping Kang, Sen Zhang, Yuchang Li, Zhiyan Zhao, Wenguang Yang, Lu Zhao, Huiyao Wang, Tao Jiang

**Affiliations:** 1State Key Laboratory of Pathogen and Biosecurity, Institute of Microbiology and Epidemiology, Academy of Military Medical Sciences, Beijing 100071, China; 18567625401@163.com (W.L.); chenyuehong.happy@163.com (Y.C.); fengye621@126.com (Y.F.); lj-pbs@163.com (J.L.); kangxiaoping@163.com (X.K.); zhangsen2260@163.com (S.Z.); liyuchang66@163.com (Y.L.); zzhiyan812@163.com (Z.Z.); ywg2106@163.com (W.Y.); zhaolu94264@163.com (L.Z.); wanghuiyao8951@163.com (H.W.); 2School of Basic Medical Sciences, Anhui Medical University, Hefei 230032, China; 3School of Public Health, Mudanjiang Medical University, Mudanjiang 157011, China

**Keywords:** human adenovirus type 55, infectious clone, reporter virus, enhanced GFP, replication-competent, homologous recombination

## Abstract

Human adenovirus 55 (HAdV-55) has recently caused outbreaks of acute respiratory disease (ARD), posing a significant public threat to civilians and military trainees. Efforts to develop antiviral inhibitors and quantify neutralizing antibodies require an experimental system to rapidly monitor viral infections, which can be achieved through the use of a plasmid that can produce an infectious virus. Here, we used a bacteria-mediated recombination approach to construct a full-length infectious cDNA clone, pAd55-FL, containing the whole genome of HadV-55. Then, the green fluorescent protein expression cassette was assembled into pAd55-FL to replace the E3 region to obtain a recombinant plasmid of pAd55-dE3-EGFP. The rescued recombinant virus rAdv55-dE3-EGFP is genetically stable and replicates similarly to the wild-type virus in cell culture. The virus rAdv55-dE3-EGFP can be used to quantify neutralizing antibody activity in sera samples, producing results in concordance with the cytopathic effect (CPE)-based microneutralization assay. Using an rAdv55-dE3-EGFP infection of A549 cells, we showed that the assay could be used for antiviral screening. Our findings suggest that the rAdv55-dE3-EGFP-based high-throughput assay provides a reliable tool for rapid neutralization testing and antiviral screening for HAdV-55.

## 1. Introduction

Human adenovirus (HAdV), first identified in 1953 [1], is an important pathogen associated with acute respiratory disease (ARD), acute gastroenteritis, epidemic keratoconjunctivitis, and genitourinary illnesses in both children and adults [2]. To date, at least 113 HAdV g have been reported and defined according to the new paradigm based on genomics, and these have been classified into seven species (A–G). Specific genotypes are often associated with particular clinical manifestations [3].

HAdV-B group infection has become an important factor in ARD [2,4,5]. Adenovirus infections are responsible for 5–10% of respiratory infections in children and 1–7% in adults, causing pneumonia in up to 20% of newborns and infants. In severe cases of HAdV pneumonia, the mortality rate may exceed 50%. In mainland China, adenovirus was detected in approximately 5.8–13% of acute respiratory tract illness (ARTI) patients, particularly in children and young adults [6,7]. HAdV-55 was first identified as an atypical HAdV-11 strain in Spain (1969), which then appeared in a Turkish military camp (2004), in Singapore (2005), and in the USA [8,9,10,11]. Now, hAdV-55 has spread widely and continues to cause outbreaks globally [9,10,12,13]. In China, HAdV-55 and HAdV-7 were identified as the leading group B adenovirus types, which cause outbreaks in military camps and schools with significantly different seasonal patterns and attack rates, based on retrospective surveillance data from 2009 to 2020 [14]. Considering that HAdVs are not included in the national notifiable infectious disease surveillance system in China and that most of the HAdV outbreaks are not identified, the prevalence of adult acute respiratory infections caused by HAdV-55 may be much higher than the actual reported cases, and its potential threat cannot be ignored.

In recent years, there have been some epidemiological analyses and serum studies on HAdV-55, all of which indicate that HAdV-55 has a high risk of transmission and a low immune background in the population [4,6,14,15]. These data highlight the importance of developing effective prevention and control strategies for adenoviruses. Active surveillance, prompt diagnosis, and the development of efficient vaccines are necessary to prevent and mitigate the spread of this virus and protect public health.

Classical cell-based antiviral assays include plaque reduction neutralization tests (PRNT), CPE-based microneutralization (MN) assays, and quantitative PCR, which are time consuming and labor intensive [1,16]. Accordingly, various rapid and reliable cell-based high-throughput assays based on reporter viruses have been developed, for viruses such as Japanese encephalitis virus, dengue virus, yellow fever virus, enterovirus type 7, and influenza virus [17,18,19]. Several recombinant HAdV-expressing reporters, such as green fluorescence protein (GFP), luciferase or alkaline phosphatase reporters, for example, Ad4-Luc, Ad7-Luc, Ad14-EGFP, and Ad14-SEAP, have been developed [20,21,22,23,24]. Increasing concern has been focused on HAdV-55-associated illnesses during the past six decades. However, although limited recombinant viruses exist in several types of HAdVs, no similar reports are available for HAdV-55. In this study, we developed and characterized a stable HAdV-55 expressing the enhanced GFP (eGFP), based on the wild-type HAdV-55 strain Y16/SX/2011 [8], which could be used for the antiviral screening and quantification of neutralizing antibodies.

## 2. Materials and Methods

### 2.1. Viruses and Cells

The human lung cancer cell line A549 (ATCC, catalog number CCL-185) and human embryonic kidney cell line 293A, a subclone of the 293 cell line (ThermoFisher, catalog number R70507), were cultured in Dulbecco’s modified Eagle’s medium (DMEM). HAdV-55 strain Y16/SX/2011 (GenBank accession number KF911353) [8] was prepared in A549 cells with Dulbecco’s modified Eagle’s medium (ThermoFisher, New York, NY, USA), supplemented with penicillin (100 IU/mL), streptomycin (100 μg/mL), and 2% fetal bovine serum (ThermoFisher, New York, USA) at 37 °C in the presence of 5% CO_2_.

### 2.2. Construction of Full-Length Infectious cDNA Clone of HAdV-55

The full-length infectious cDNA plasmid pAd55-FL harboring the HAdV-55 (Y16/SX/2011) genome was constructed with a previously described strategy [8] using the highly efficient homologous recombination method in *E. coli* BJ5183 (TAKARA, Dalian, China) (Figure 1). In brief, the Amp-Ori fragment was amplified from the pAdEasy-1 plasmid (Beyotime, Shanghai, China) with the primers P617-AdBackbone-F-EcoRI and P618-AdBackbone-R-HindIII (Table 1), then ligated with a multiple cloning site containing restriction sites (*Hind* III, *Pme* I, *EcoR* I, etc.) to constitute the pAdBone. The PCR-generated homologous regions HAL/HAR to HAdV-55 were cloned into pAdBone as an *Asis* I-HAL-*Pme* I- HAR-*Asis* I fragment to construct the rescued plasmid pAdBone-LRA, which then was linearized using the restriction enzyme *Pme* I. The gel-purified linearized pAdBoneLRA were mixed with HAdV-55 genomic DNA and transformed into *E. coli* BJ5183 competent cells (TAKARA, Dalian, China) by electroporation. The positive full-length infectious cDNA pAd55-FL was detected by colony PCR. The pAd55-FL was analyzed with restriction enzymes *Asis* I and *Nde* I by gel electrophoresis, and then sequence verified using Sanger sequencing.

### 2.3. Constructure of the E3-Defective Replication-Competent EGFP Expression Vector pAd55-dE3-EGFP

The plasmid pAd55-dE3-EGFP was constructed as shown in Figure 1. First, the homologous regions E3L/E3R to the E3 gene were amplified with primers E3L1000-F/E3L1000-R and E3R1000-F/E3R1000-R, and ligated by Gibson DNA ligase into pEGFP-N1 (Clonetech, Moutain View, USA) to construct the shuttle vector pEGFP-N1-E3LR containing the homology arms. After *Avr* II digestion, the plasmid pAd55-FL was recovered from the agarose gel, and then transformed into BJ5183 chemically competent cells (TAKARA, China) with *Mlu* I-digested shuttle plasmid pEGFP-E3LR to construct plasmid pAd55-dE3-EGFP. The EGFP was introduced into the plasmid in place of the whole E3 gene by homologous recombination between E3L and E3R fragments. The recombinant plasmid was selected by colony PCR using insert-specific primers E3D-Det-F2 and AVRII-R and then verified by *Ned* I restriction enzyme analysis and DNA sequencing.

### 2.4. Virus Rescue

The plasmids pAd55-FL and pAd55-dE3-EGFP were digested with *Asis* I to release the viral genomic DNA, which was transfected into 293A cells using Lipofectamine 3000 (ThermoFisher, Carlsbad, USA) according to the manufacturer’s instructions. The transfected cells were cultured for 6 to 10 days until the CPE was observed. Progeny viruses designated rAd55 and rAd55-dE3-EGFP that were collected from the supernatant were termed passage 0 viruses and were used to produce stocks. The virus stocks were prepared in A549 cells. Once CPE was observed, the culture media were repeatedly freeze–thawed 3 times, and all cell lysates were collected and centrifuged at 10,000× *g* for 10 min to remove the impurities and precipitates. The supernatant was harvested and titered for subsequent analysis. The 50% tissue culture infectious dose (TCID_50_) mL^−1^ titer in A549 cells was determined using the Reed–Muench method.

### 2.5. Virus Growth Kinetics in A549 Cells

The virus growth kinetics were observed in A549 cells. In brief, 24 h before infection with HAdV-55, rAd55, and rAd55-dE3-EGFP, A549 cells were seeded in 24-well plates at a density of 2 × 10^5^ cells per well. Cells were washed three times with PBS and inoculated with viruses (MOI = 0.05 TCID_50_ units/cell). After 2 h incubation, the supernatant was removed and cells were washed three times with PBS and supplied with the medium as described above. Supernatants and cells were collected at 2, 24, 48, 72, 96, and 120 h post-infection separately. Viral genomic DNA was extracted with a TaKaRa MiniBEST Viral RNA/DNA Extraction Kit Ver.5.0 (Takara, Dalian, China) according to the manufacturer’s instructions. The replication of viral DNA in cells was measured by qPCR with adenovirus-specific primers qHAdV-UniF/-UniR and probes qHAdV-UniProbe (Table 1). The viral genome copy number based on the plasmid standard curve method was calculated and plotted using GraphPad Prism 9 (GraphPad Software Inc., San Diego, CA, USA). The TCID_50_ mL^−1^ titer in A549 cells was determined using the Reed–Muench method.

### 2.6. Stability Testing of rAd55-dE3-EGFP Containing eGFP and Viral Genome in A549

The virus rAd55-dE3-EGFP was serially passaged in A549 cells twenty times at an MOI of 1 to obtain the P0-P20 viruses. The eGFP expression was observed at 24 hpi under the fluorescence microscope. The inserted eGFP genes in serially passaged rAd55-dE3-EGFP viruses were identified by gel electrophoresis of eGFP PCR products (EGFP-F/EGFP-R) and sequencing analysis.

### 2.7. Indirect Immunofluorescence and Fluorescence Focus Assay

The A549 cells grown in a 12-well or 96-well plate were infected with HAdV-55, rAd55, and rAd55-dE3-EGFP, respectively. The viral culture was 10-fold diluted in series. The infected cells were cultured for 40–48 h in culture at 37 °C and 5% CO_2_ and then we removed the medium. The cells were fixed with cold methanol at −20 °C for 10 min. After 30 min blocking with PBS containing 1% bovine serum albumin (BSA-PBS), the cells were incubated with hexon-specific antibody (B025/AD51, abcam, UK) at 37 °C for 1 h, then with a CY3-conjugated goat anti-mouse secondary antibody (Servicebio, Wuhan, China) for another 1 h. Then, 40, 60-diamidino-2-phenylindole (DAPI, Servicebio, China) was used for nuclear staining. The cells were observed and photographed under a fluorescence microscope at the excitation/emission wavelengths of 470 nm/515 nm, respectively.

### 2.8. Microneutralization Assay

The titers of neutralizing antibodies (nAbs) against HAdV-55 were determined using the Reed–Muench formula as previously described [25] or Ad55-dE3-EGFP-based MN assay. In brief, the A549 cells were seeded into 96-well plates at 2 × 10^4^ cells per well. After 24 h, serial dilutions of the mouse serum were incubated with rAd55 at 100 TCID_50_ particles per well for 1 h at 37 °C, then added to the 96-well plates and incubated at 37 °C. CPE was analyzed at 120 h. For the eGFP-based MN assay, the serum was incubated with rAd55-dE3-EGFP. At 24 h post-infection, fluorescence images were collected from three random fields at the same exposure intensity and position in each well. The fluorescence intensity was quantified by microscope built-in software. Relative light units (RLUs) were recorded and the nAbs titers were calculated as the dilutions that inhibited 50% RLU values.

### 2.9. Antiviral Drug Screening

The A549 cells were seeded at a density of 10^4^ cells per well in a 96-well plate, incubated for 24 h, and then infected with rHAd55 or recombinant virus rHAd55-dE3-EGFP at an MOI of 0.5 (TCID_50_ units/cell) and treated with varying concentrations ranging from 0.00128 to 4 mM of cidofovir or brincidofovir for 24 h, followed by observation of eGFP expression under a fluorescence microscope. The Image J software was employed to process the images and calculate the overall fluorescence value for each picture, from which the fluorescence inhibition ratio was determined. The IC_50_ value of green fluorescence inhibition was calculated using GraphPad Prism 9.0 software (GraphPad Software) and representing the minimum effective concentration of the drugs.

### 2.10. Electron Microscopic Observation

The virus was prepared in A549 cells. Once CPE observed, the culture media were repeatedly freeze–thawed 3 times, and all cell lysates were collected and centrifuged at 10,000× *g* for 1 h to remove the impurities and precipitates. After another centrifugation at 50,000× *g* for 1 h, the precipitate was resuspended in 1 mL PBS + 4 mL CsCl (45.4%), and then centrifugated at 180,000× *g* for 12 h. The pellet was resuspended at 20 µL PBS (pH 7.2). The CsCl-purified virus was stained with phosphotungstate acid and observed under the transmission electron microscope (JEOL, Tokyo, Japan).

### 2.11. Mouse Experiment

Female BALB/c mice (8 weeks old, five mice per group) were intramuscularly immunized with HAdV-55, rAd55, rAd55-dE3-EGFP, or PBS once every 2 weeks a total of 4 times. Mouse serum was collected two weeks after the last immunization.

### 2.12. Statistical Analysis

Data are presented as means ± SD of at least three independent experiments. Differences between the two groups were evaluated using Student’s *t*-test, and differences between three or more groups were evaluated using analysis of variance (ANOVA). Statistical analysis was performed using GraphPad Prism 9 (GraphPad Software). Statistical significance was set at *p* < 0.05.

## 3. Results

### 3.1. Generation of the Infectious Clones pAd55-FL/pAd55-dE3-EGFP

The plasmid pAd55-FL was obtained in *E. coli* BJ5183 by the homologous recombination of linear HAdV-55 viral genomic DNA and the shuttle vector pAdBoneLRA (Figure 1 and Section 2.2 and Section 2.3). The plasmid pAd55-dE3-EGFP was obtained by homologous recombination in *E. coli* of *Avr* II-linearized pAd55-FL and *Mlu* I-digested shuttle vector pEGFP-E3LRA. The recombinant plasmids pAd55-FL and pAd55-dE3-EGFP were confirmed by restriction enzyme *Nde* I digestion (Figure 2a). The plasmid pAd55-FL was identified by restriction enzyme *Nhe* I digestion, resulting in four fragments of 15,189 bp, 10,369 bp, 7045 bp, and 5815 bp. The plasmid pAd55-dE3-EGFP was confirmed by restriction enzyme *Nde* I digestion, generating five fragments of 13,000 bp, 8500 bp, 7600 bp, 4000 bp, and 3200 bp, as observed on agarose gel electrophoresis (Figure 2a). Two plasmids, pAd55-dE3-EGFP and pAd55-FL, were amplified using E3D-Det-F and AVRII-R primer pairs, with PCR products of 2500 bp and 4200 bp, respectively (Figure 2b).

### 3.2. rAd55 and rAd55-dE3-EGFP Rescue

The plasmids pAd55-FL and pAd55-dE3-EGFP were linearized with *Asis* I and then transfected the 293A cells. At about 7 days post-transfection, the CPE were significant and the rescued recombinant viruses rAd55 and rAd55-dE3-EGFP were harvested and then inoculated into A549 cells. Obvious CPE was observed 48 h post-infection (hpi) with both viruses (Figure 3a). In the cells infected with rAd55-dE3-EGFP, EGFP was observed as early as 24 hpi and increased over time, while CPE became apparent at 96 hpi (Figure 3b).

### 3.3. Stability Testing of rAd55-dE3-EGFP

The genetic stability of the recombinant reporter virus rAd55-dE3-EGFP was assessed by serial passage on A549 cells. The P1, P6, P12, or P20 virus was titered and incubated into A549 cells at 1 MOI. EGFP and CPE were observed at 24 hpi and increased over time (Figure 4a). After the nucleic acid extraction from the viruses, the eGFP gene was identified by PCR amplification and nucleic acid electrophoresis and sequencing. The results showed that there was no loss and no mutation in the EGFP gene over serial passages (Figure 4b).

### 3.4. Growth Kinetics Characteristics of rAd55 and rAd55-dE3-EGFP

As shown in Figure 5, the peak of DNA replication for both viruses was reached at approximately 48 hpi, while the peak of viral reproduction was reached at 120 hpi. There was no significant difference in the replication kinetics between HAdV-55, rAd55, and rAd55-dE3-EGFP. The relative fluorescence of the rAd55-dE3-EGFP viruses increased over time, but fluorescence decreased once CPE was observed (Figure 5c). The data showed similar titers among HAdV-55, rAd55, and rAd55-dE3-EGFP after 120 hpi. The viruses increased to 5 × 10^5^ TCID_50_/mL at 48 h post-infection and up to 5 × 10^7^ TCID_50_/mL by 120 hpi. Both rAd55 and rAd55-dE3-EGFP had DNA replication efficiency that was very similar to that of the parental HAdV-55. The results, depicted in one-step growth curves, showed that both rAd55 and rAd55-dE3-EGFP had DNA replication efficiency that was very similar to the parental HAdV-55.

### 3.5. Indirect Immunofluorescence Assay

An indirect immunofluorescence assay was performed to determine where the viral replication and hexon gene expression of HAdV-55, rAd55, and rAd55-dE3-EGFP viruses were present. The A549 cells were infected with HAdV-55, rAd55, and rAd55-dE3-EGFP, respectively. As shown in Figure 6, the red fluorescence labeled by CY3 and green fluorescence from rAd55-dE3-EGFP was observed. It can be seen that the position of the green fluorescence corresponds to the locations of the CPE and that the green and red fluorescent areas overlapped significantly. Using DAPI to label the nucleus, we can see that the green and red fluorescence and the blue DAPI were also located in the same regions. At the same time, based on the red fluorescence, there was no difference between the hexon protein expression of the recombinant virus and the wild-type strain virus.

### 3.6. Development of rAd55-dE3-EGFP-Based MN Assays

Fifteen positive mouse antisera were tested for neutralization titer against HAdV-55 using an rAd55-dE3-EGFP based MN assay. We took green fluorescence images and numerically recorded the fluorescence intensity in the field of view at the same position at 24 hpi. The CPE- and the eGFP-based MN assay agrees with the positive sera, but the eGFP-based MN assay has slightly higher antibody titers than the CPE-based method (Figure 7a). As shown in Figure 7b, the inhibition of green fluorescence by the positive sera was obvious at 24 hpi, as the neutralization potency increased. This is probably because the CPE-based method requires CPE to determine the observable degree, whereas eGFP fluorescence can be detected by eGFP expression without determination by the presence of CPE. This result shows that the viral replication could be determined by eGFP expression, which shortened the endpoint observation time of the MN assay to as early as 24 hpi.

### 3.7. Antiviral Screening Experiment

As seen in Figure 8, both cidofovir and brincidofovir showed good inhibitory activity against rAd55-dE3-EGFP, but brincidofovir exhibited a stronger antiviral effect than cidofovir; at a concentration of 0.0064 μM, significant inhibition of green fluorescence expression was observed. We used high-content fluorescence microscopy to capture images of all cell wells under the same exposure conditions. Image J software was used to process the images and calculate the overall fluorescence value for each picture. The fluorescence inhibition ratio was calculated to obtain IC_50_ values using GraphPad Prism 9.0 software. The IC_50_ of brincidofovir and cidofovir was 0.01785 μM and 0.3604 μM, respectively, which is consistent with the published studies reporting IC_50_ values for brincidofovir of 0.009–0.28 μM and for cidofovir of 0.5–6.2 μM [26]. The results indicated that the EGFP reporter-virus-based antiviral screening method is a reliable and efficient means for evaluating drug efficacy against adenovirus.

### 3.8. Electron Microscopic Observation

The negatively stained viruses were observed using transmission electron microscopy. As shown in Figure 9, symmetric hexagonal particles were observed, indicating a symmetric icosahedral structure, which is consistent with the typical particle morphology of HAdV. The diameters of all three virus particles observed under the electron microscope were approximately 70–90 nm. These results indicate that HAdV-55, rAd55, and rAd55-dE3-EGFP have similar morphological characteristics typical of HAdVs.

## 4. Discussion

HAdV-55 re-emerged in 2005 and has caused more ARD outbreaks with severe and fatal pneumonia in people of all ages in Singapore [27], the United States [15,28], Turkey [29], Pakistan, South Korea, and Israel [2,4,30], which has generated extensive concern. Given that the pre-existing immunity against HAdV-55 in the population is unclear, it is important to clarify the circulation of HAdV-55 nAbs in the general population. In addition, the development of vaccines and drugs against viruses is of great importance to human health. No antiviral compounds or vaccines have yet been approved for the treatment and prevention of HAdV-55 infections. As a result, the construction of a stable EGFP-expressing HAdV-55 reporter virus will facilitate further study of gene functions, pathogenic mechanisms, and recombination machinery, as well as the development of vaccines and antiviral agents against adenoviruses.

The key challenge for the construction of a replication-competent infectious clone of adenovirus is that the genome is as large as 35 kb and has a small appropriate restriction enzyme cutting site [31]. Currently, the most common strategy for large-genome viruses is to utilize homologous recombination in bacteria, such as pAdeasy [32,33]. Recombinant plasmids lacking the E3 region were constructed through incomplete restriction digestion and homologous recombination within bacteria [34]. However, that method required extensive time for screening and determining positive bacterial clones, and the repeatability was found to be unreliable [22]. Based on the fusion PCR method, the infectious clones of HAdV-14 and HAdV-55 were constructed; however, mutations, insertions, or deletions during multiple long-fragment PCR amplifications were problematic [35,36,37]. The method developed in the current study for adenovirus recombination offers several advantages over other reported methods. It avoids the need for an intermediate plasmid and simplifies the construction of the recombinant plasmid. The homologous recombination and seamless cloning ensure both efficiency and accuracy. Moreover, the current method without the requirement of pAdeasy strains prevented unintended mutations during plasmid replication in recombinase-containing bacteria, and reduced the occurrence of false positives in monoclonal colonies [33,35,36,37,38].

The absence of the E1 gene can affect the biological properties of the adenoviruses [34,39]. The E3 gene is a non-essential region for adenovirus replication in cell culture. The deletion of the E3 region does not affect viral replication and proliferation, while the insertion of the transgene into the E3 14.7K region produces a high level of tumor-selective transgene expression [40]. The E3 region is an early expressed protein, and foreign genes inserted into the E3 region are more likely to be highly expressed [41]. In our study, the E3 region was deleted and the exogenous EGFP gene was inserted, which did not affect the biological properties of HAdV-55 and was more consistent with the wild-type virus in drug screening. Meanwhile, the E3 region is subject to the control of early promoters and can be expressed at the early stage of viral replication, which can shorten the observation window [42]. Additionally, by comparing the consistency in titeration of the neutralizing antibody, we found that the antibody titers detected by the reporter-virus-based method were higher than those obtained by the traditional CPE-based MN assay, which is in agreement with the published studies [22,24,43]. Overall, both methods produced similar results in this study, which indicates that the method of detecting serum-neutralizing antibodies with a recombinant reporter virus is more sensitive and has a shorter observation time.

In summary, a full-length infectious genomic clone and a stable EGFP-expressing recombinant clone were successfully constructed, whose biological characteristics were as expected. The replication-competent infectious clones may facilitate and provide powerful tools for adenovirus pathogenicity, the development of vaccines or antiviral screening and antibody serological investigation, and a strategy for the development of other respiratory adenovirus reporter systems.

## Figures and Tables

**Figure 1 viruses-15-01192-f001:**
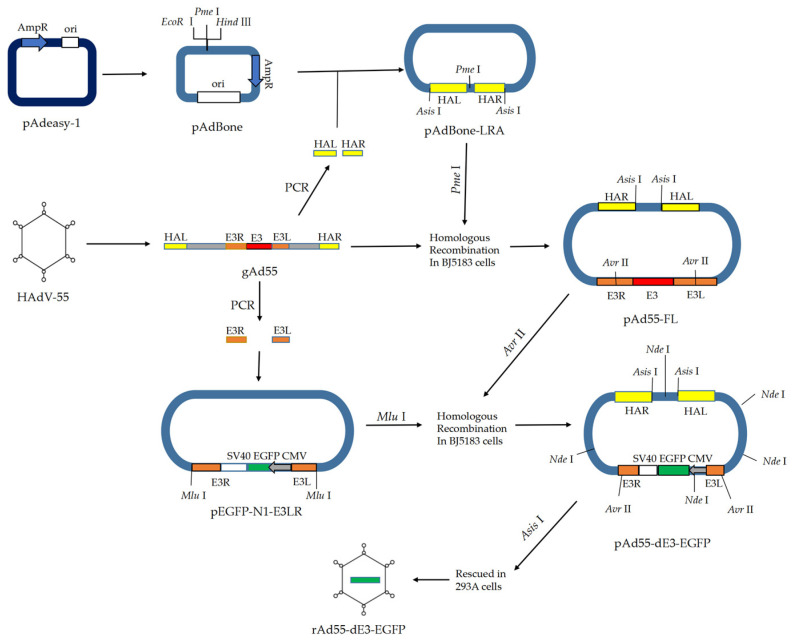
Schematic representation of the construction of pAd55-FL and pAd55-dE3-EGFP. AmpR: ampicillin resistance gene; CMV: CMV promotor; E3: E3 gene of HAdV-55; E3L and E3R: left and right homologous arm to the E3 gene; EGFP: enhanced green fluorescence protein; gAd55: genome of HAdV-55 virus; *Hind* III, *Pme* I, *EcoR* I, *Avr* II, *Asis* I and *Nde* I represent restriction sites; HLA and HRA: left and right homologous arm to the virus genome of HAdV-55; ori: pBR322 origin of replication; pEGFP-N1: vector for fusing EGFP to the C-terminus of a partner protein; pAdBone-LRA: recombinant pAdBone with HAL and HAR homologous arm regions; pAd55-FL: plasmid harboring the full-length cDNA of HAdV-55 virus genome; rAd55-dE3-EGFP: recued HAdV-55 virus encoding EGFP without E3 gene.

**Figure 2 viruses-15-01192-f002:**
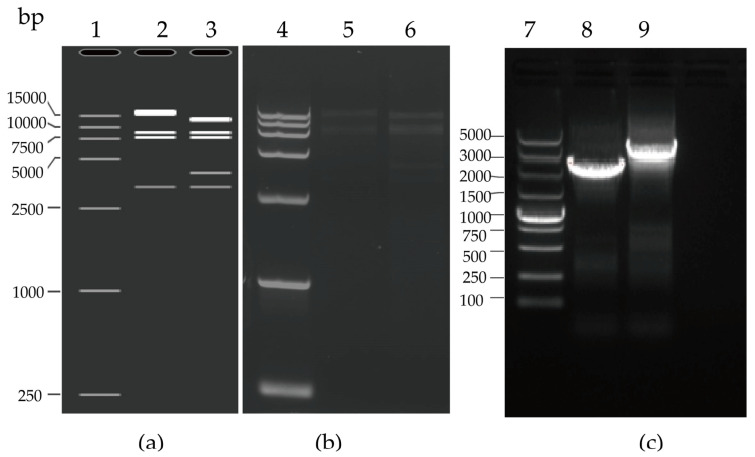
Restriction enzyme analysis and PCR confirmation of recombinant plasmids pAd55-FL and pAd55-dE3-EGFP. (**a**) In silico restriction map generated by SnapGene. Lane 1: DL 15,000 DNA marker (TAKARA, China); Lane 2: pAd55-FL/*Nde* I; Lane 3: pAd55-dE3-EGFP/*Nde* I; (**b**) Restriction map of pAd5-FL and pAD55-dE3-EGFP. Lane 4: DL 15,000 DNA marker (TAKARA, China); Lane 5: pAd55-FL/*Nde* I; Lane 6: pAd55-dE3-EGFP/*Nde* I; (**c**) PCR amplification with E3D-Det-F2/AVRII-R. Lane 7: DL5000 DNA marker (TAKARA, China); Lane 8: PCR amplification with E3D-Det-F2/AVRII-R of pAd55-dE3-EGFP; Lane 9: PCR amplification with E3D-Det-F2/AVRII-R of pAd55-FL.

**Figure 3 viruses-15-01192-f003:**
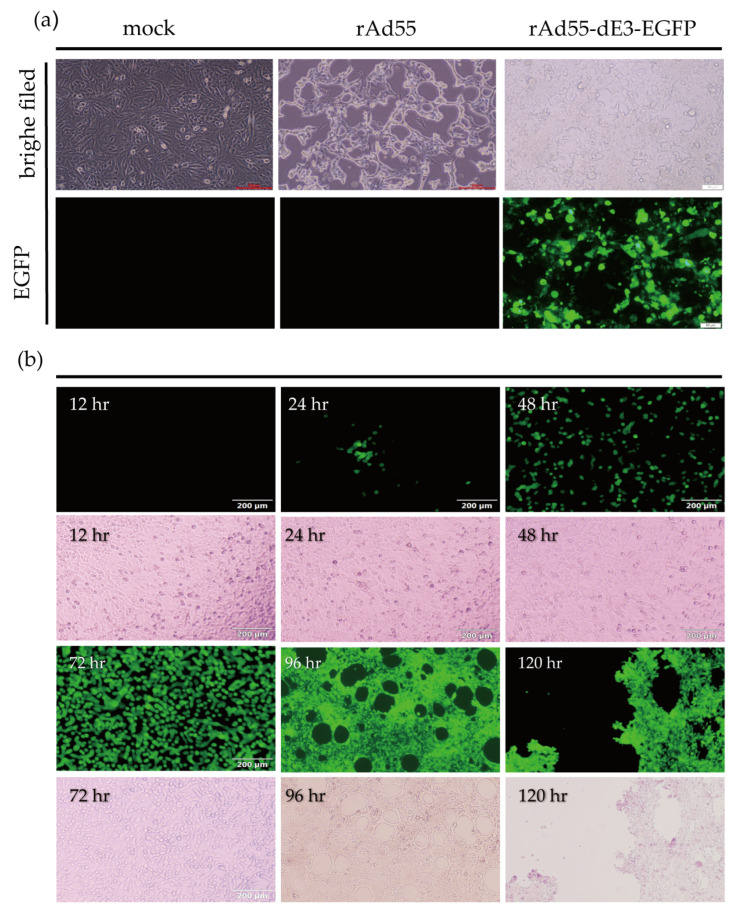
rAd55 and rAd55-dE3-EGFP rescue. (**a**) CPE and EGFP expression of rAd55 and rAd55-dE3-EGFP at 7 days post-transfection, scale bar: 50 μm. (**b**) EGFP expression of rAd55-dE3-EGFP at indicated time. Visualized by fluorescence microscopy under fluorescent light and visible light, scale bar: 100 µm.

**Figure 4 viruses-15-01192-f004:**
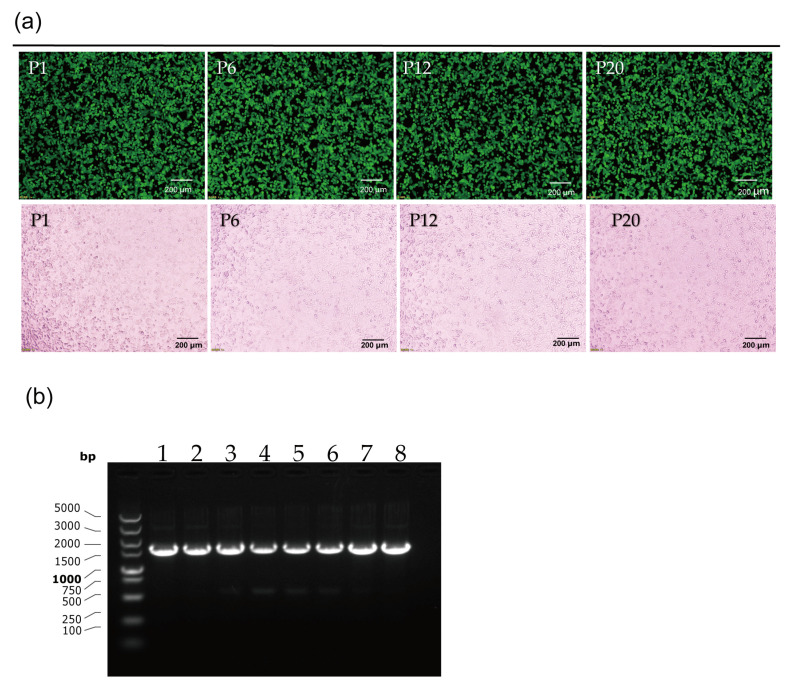
Stability testing of rAd55-dE3-EGFP. (**a**) EGFP expression and CPE in cells infected with P1, P6, P12, and P20 viruses under fluorescent or visible light. (**b**) The EGFP gene identified by nucleic acid electrophoresis. Scale bar: 200 um.

**Figure 5 viruses-15-01192-f005:**
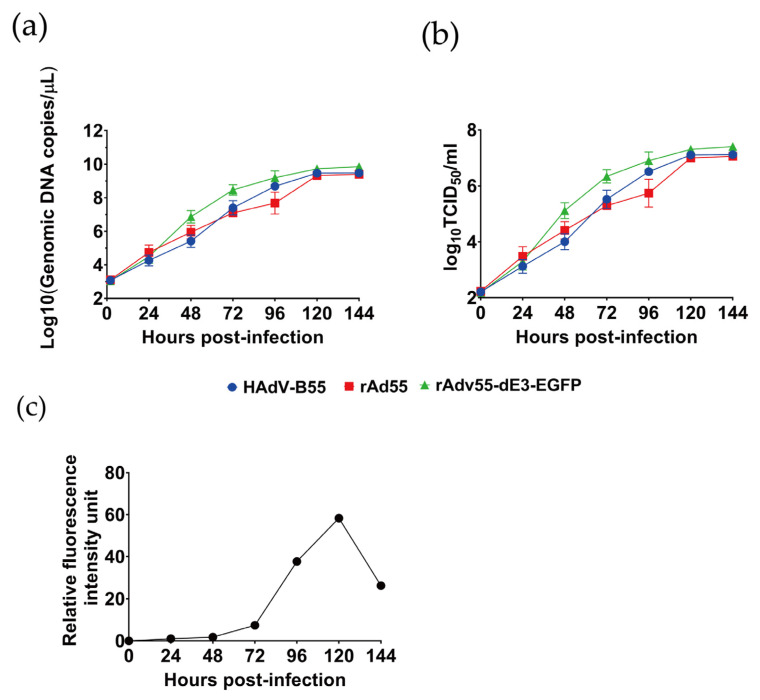
Growth kinetics characteristics of rAd55 and rAd55-dE3-EGFP. (**a**) The viral growth of HAdV-55, rAd55, and rAd55-dE3-EGFP viruses in A549 cells. (**b**) Viral DNA replication of HAdV-55, rAd55, and rAd55-dE3-EGFP viruses in A549 cells. (**c**) Relative fluorescence of rAd55-dE3-EGFP virus.

**Figure 6 viruses-15-01192-f006:**
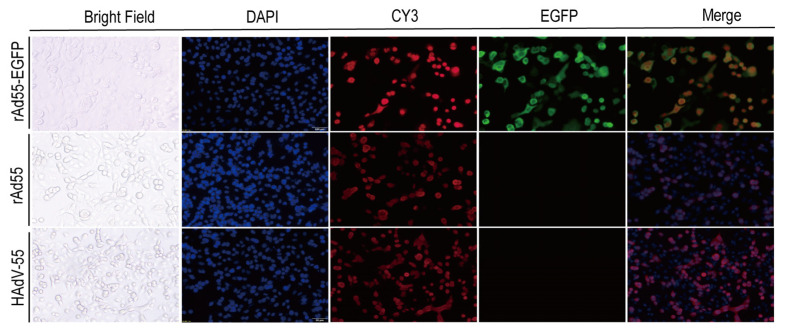
Expression pattern of EGFP in virus-infected A549 cells. A549 cells were infected with HAdV-55, rAd55, and rAd55-dE3-EGFP viruses. Red: hexon, blue: DAPI nuclear stain, green: EGFP. Scale bar: 50 µm.

**Figure 7 viruses-15-01192-f007:**
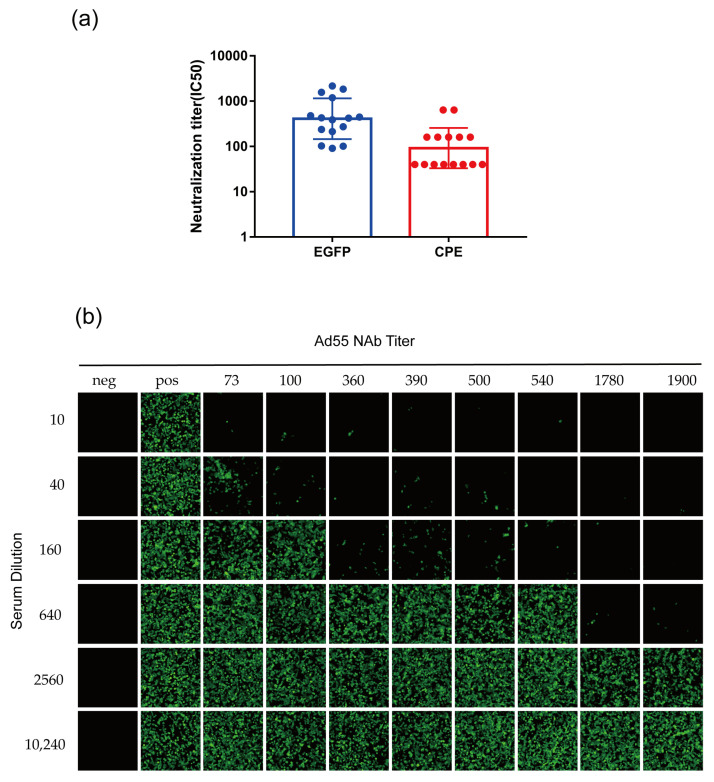
The validation of MN assays based on rAd55-dE3-EGFP. (**a**) Mouse antisera nAb titers were determined using CPE-based MN assays (red dots) and eGFP-based MN assays (blue dots), respectively. (**b**) Serial dilutions of HAdV-55-seropositive sera with nAb titers <200, 200–500, 500–1500, and >1500 were incubated with rAd55-dE3-EGFP and then infected A549 cells. Green: eGFP. Scale bar: 100 µm.

**Figure 8 viruses-15-01192-f008:**
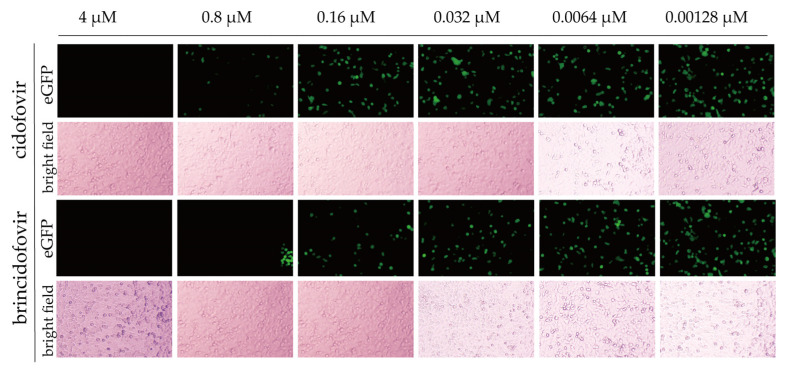
Antiviral activities of cidofovir and brincidofovir on rAd55-dE3-EGFP viruses. Green: eGFP. Scale bar: 100 µm.

**Figure 9 viruses-15-01192-f009:**
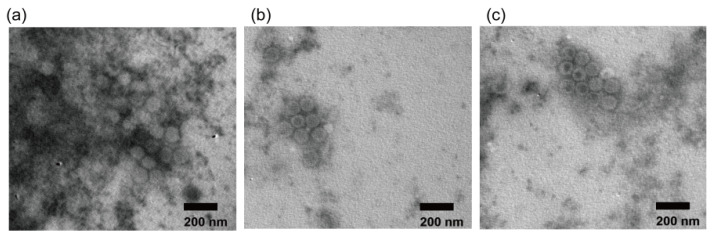
Negative-stained EM images of adenoviruses. (**a**) HAdV-55; (**b**) rAd55; (**c**) rAd55-dE3-EGFP. Scale bar: 200 nm.

**Table 1 viruses-15-01192-t001:** Primers used in the study.

Plasmid	Primer	Sequence (5′-3′) #
pAd55	P762-LARM-HindIII-AsiSI-F	gcgAAGCTTGCGATCGCcatcatcaataatataccttatag
	P606-LARM-PmeI-R	ataGTTTAAACgtcttcatagcagtgcaaatcacag
	P607-RARM-PmeI-F	ataGTTTAAACcaccagtaatgtcatcaaagttgctg
	P763-RARM-EcoRI-AsiSI-R	ataGAATTCGCGATCGCcatcatcaataatataccttatag
pEGFP-E3LR	E3L1000-F:	*CGTATTACCGCCATGCATT*AGTTATTAATACGCGTctggttagctgcgcagccggcatc
	E3L1000-R:	*CTAATGACCCCGTAA*TTGATTACTATTAATgcagtggtctaaatgtcgcagccgag
	E3R1000-F:	GTTTGTCCAAACTCATCAATGTATCTTAAGatgcggactaagagacctgctac
	E3R1000-R:	*AATATTAACGCTTACAAT*TTACGCCTTAAGACGCGTgatgagacacaatcgcccgatcc
PCR	E3D-Det-F2	cacccctcgtcagactgttttgac
	AVRII-R	cactggagtccatcatttgacag
qPCR	qHAdV-UniF	ATGGCCACCCCATCGAT
	qHAdV-UniR	ACTCAGGTACTCCGAAGCATCCT
	qHAdV-UniProbe	FAM-TGGGCATACATGCACATCGCCG-BHQ1
pAdBone0	P617-AdBackbone-F-EcoRI	ataGAATTCTCGACCGATGCCCTTGAGAGCCTTC
	P618-AdBackbone-R-HindIII:	gcgAAGCTTCAGGTGGCACTTTTCGGGGAAATG
EGFP	EGFP-F	cgttacataacttacggtaaatgg
	EGFP-R	taagatacattgatgagtttggacaaac

# Uppercase letters are restriction enzyme sites, italic letters are vector homologous recombination sequences, and lowercase letters are viral sequences. The letters underlined are extra bases added on to either side of the recognition site to cleave efficiently).

## Data Availability

Not applicable.

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
