# Peer review of "Generation and Characterization of a Replication-Competent Human Adenovirus Type 55 Encoding EGFP"

_viruses, 2023, doi:10.3390/v15051192_

Round 1

Reviewer 1 Report

This manuscript describes creation of a mutant Had55 virus that is deleted for early region 3 (E3) and expresses the eukaryotic green fluorescent protein (EGFP). After creating the virus, the authors do some characterization of it that demonstrates its utility for testing antivirals and neutralizing antibody responses. Although many of the methods are fairly well documented, others are missing. I think the key figure for the method is Fig. 1 and it could be arranged better and with bigger fonts to make it clearer. I roughly sketched out how it could be rearranged and have included that figure.  There are also many copyediting errors that I just marked directly on a copy of the manuscript, to save time.

Some important points that should be addressed:

1. Line 16. “reverse genetic system” is a term misappropriated from genetics by virologists. It is better to talk about a system for making an infectious clone.  Thus, the sentence could be modified, “which can be achieved through use of a bacmid that can produced infectious virus.”

2. Line 75 and line 214. What are 293A cells? The ATCC reference only is for 293 cells, the original parental line isolated by Frank Graham.

3. Figure 1. In addition to comments on the copyedited manuscript and the diagram, the legend needs to include more explanation, for example, what are some of the abbreviations used in the figure (HAR, HAL, Ant I, II, etc.)?  A key thing for this figure will be to include the relevant restriction sites, NdeI and Asis I on the appropriate plasmids and virus genomes.

4. Where does the EGFP gene end up in the final vectors? Is it in place of the E3 gene, or somewhere else in the virus? That should be clearly specified.

5. Line 87. BJ5183 cells are not described in reference 8, and a reference (or vendor/source) should be provided. 

6. Line 120. How were stocks prepared? Are they lysates, clarified after freeze-thaw 3x? Are they CsCl purified? This becomes important for the subsequent discussion in line 174.  How the viruses were titrated should be described here too (around line 120), not 15 lines later in lines 134-135, where a TCID50 assay is described. Were plaque assays also done, and if so on what cells? PFU are listed in Fig. 5A y-axis…

7. Line 126. The MOI should have the units listed:  (MOI = 0.05 TCID50 units/cell).

8. Line 119, 144. rAd55: is this the same as what is generated from pAd55-FL?  Does FL stand for full-length? If so, that would be an important abbreviation to explain in the legend to Fig. 1. 

9. Line 174. “Virus samples were collected…”. What does this mean? Are these cell lysates of CsCl purified virus, or ???

10. Line 176. What is the concentration of CsCl?

11. Line 181. What is the strain, sex, and age of the mice?

12. Figure 2 needs a lot of work.  I think the first three lanes are in silico SnapGene analysis. Then I think another figure is put right up against it, lanes 4-6. It appears to be an actual agarose gel. It should have white space between lanes 3 and 4 – the in silico analysis and the actual gel image.  The bands in lane 4 in particular have wings and dimples that make it look like an actual sample electrophoresed on agarose.  The legend doesn’t describe what lanes 4 and 5 are!   Lane 1 appears to be some kind of ladder (“DL15000bp”) – what is the source of this? Lanes 5 and 6 are faint -- can that figure be improved so the bands will show? The legend describes a lane 7 for Fig. 2a but lane 7 is in Fig. 2b. Fig. 2b, lane 7 appears to be a ladder – what is that?  The legend describes two lane 9s. Altogether, this makes this figure uninterpretable and unconvincing.

13. Lines 229 and 231 describe Figs. 2d and 2e. There are no such figures!

14. Fig. 5. The y-axis for 5a and b is not Llog10 data. The data are shown as 104, 105, etc. If the log10 data were shown, it would be 4, 5 etc.

15. Fig. 6 needs a lot of work in the explanation section starting around line 258. It should be stated right here that the red staining is a Cy-5 labeled antibody to hexon (source? reference?). Is this a directly conjugated Ab or is it detected with a Cy-5 secondary antibody? There is nothing in M&M about this experiment! 

16. Lines 260, 279. What do you mean by cell lesions? This isn’t standard terminology to me. 

17. As noted on the copyedited draft, the figure numbers for Figs. 8 and 9 are wrong. 

18. Line 329. Please clarify what is meant by “less of a restriction site.”

19. Lines 358-373. This part of the Discussion is confusing. What does all this description of other viruses have to do with the experiments here? 

Reviewer 2 Report

Manuscript by Li et al describes a generation of the Adv55-dE3-EGFP virus and its detailed characterization. As such the manuscript is well written, data is clearly presented and without a doubt, it has an impact on HAdV-55 research.

I do not have comments regarding the scientific part of the manuscript as it is a clear cut in my mind. However, I do have some concerns regarding the writting of the manuscript and not being consistent (t..ex, is it HAdV or hAdV or HadV?). The authors should check the whole manuscript for these errors as they take down the pleasure of reading it.

See above
